


# Radon-222 monitoring at German ICOS atmosphere stations

Maksym Gachkivskyi[1,2], Ute Karstens[3], Bernd Fischer[4], Dagmar Kubistin[5], Jennifer Müller-Williams[5], Matthias Lindauer[5], and Ingeborg Levin[1,†]

[1]Institut für Umweltphysik, Heidelberg University, INF 229, 69120 Heidelberg, Germany
[2]ICOS Central Radiocarbon Laboratory, Heidelberg University, Berliner Straße 53, 69120 Heidelberg, Germany
[3]ICOS ERIC Carbon Portal, Physical Geography and Ecosystem Science, Lund University, Sölvegatan 12, 22362 Lund, Sweden
[4]Umweltbundesamt, Meßstelle Schauinsland, Schauinslandweg 2, 79254 Oberried, Germany
[5]Deutscher Wetterdienst, Meteorological Observatory Hohenpeißenberg, Albin-Schwaiger-Weg 10, 82383 Hohenpeißenberg, Germany
[†]deceased, 10 February 2024

**Correspondence:** Maksym Gachkivskyi (mgach@iup.uni-heidelberg.de)

**Abstract.**

Atmospheric measurements of the short-lived radioactive noble gas $^{222}$Radon ($^{222}$Rn) have many applications. Its concentrations are driven by atmospheric mixing as well as seasonal variations, which follow the seasonality of $^{222}$Rn exhalation from continental soils with lower values during the wet winter and spring months compared to more dry summer conditions. Hence, it can be used as tracer to distinguish marine from continental air masses or for transport model validation. The Heidelberg Radon Monitor (HRM) is a static filter detector measuring atmospheric $^{214}$Polonium ($^{214}$Po), which is a progeny of $^{222}$Rn. These measurements can be used to infer atmospheric $^{222}$Rn activity concentrations if the radioactive disequilibrium between $^{214}$Po and $^{222}$Rn at the measurement site is known. In this study, $^{214}$Po activity concentrations measured with the HRM at 8 stations in the ICOS Germany network are presented, along with guidelines for evaluating these data to estimate atmospheric $^{222}$Rn activity concentrations. In addition to the established line-loss and disequilibrium corrections applied when sampling through long tubing or from air intake heights close to the ground, respectively, an upper limit for relative humidity (RH) is suggested, where secular equilibrium can still be assumed. At higher RH, aerosol scavenging effects can cause disequilibrium between $^{214}$Po and $^{222}$Rn. Using comparison with the model this threshold is determined to be at about 98% RH. A clear diurnal cycle of $^{222}$Rn is observed at all German tower stations during the summer and autumn months as well as seasonal cycle with maximum during summer and autumn months. Overall, our results demonstrate that the $^{214}$Po-based $^{222}$Rn measurements with the HRM are reliable if the equilibrium conditions between $^{214}$Po and $^{222}$Rn can be ensured, i.e. for air intake heights above 80-90 m a.g.l. at stations located in flat areas during conditions with RH < 98%.

## 1 Introduction

$^{222}$Radon ($^{222}$Rn) is a short-lived radioactive noble gas (half-life time $T_{1/2}$ = 3.8 days) and it is a progeny of $^{226}$Radium ($^{226}$Ra), which is a member of the primordial $^{238}$Uranium decay series. It is naturally produced in all soils and as gaseous constituent it can escape from the soil air into the atmosphere. The exhalation rate of $^{222}$Rn from continental surfaces depends





on the soil properties, such as grain size distribution, porosity and $^{226}$Ra content, and it varies with soil moisture and other meteorological parameters (e.g. Nazaroff, 1992; Karstens et al., 2015). The $^{222}$Rn flux from (ocean) water surfaces is negligible (Schery and Huang, 2004) compared to that from continental surfaces. The half-life time of $^{222}$Rn is sufficiently long that it can accumulate in air masses travelling over the continents, and short enough to observe a vertical activity concentration gradient with height above ground. Therefore, $^{222}$Rn can serve as tracer to distinguish continental from marine air masses (e.g. Dörr et al., 1983; Polian, 1986; Biraud, 2000). If the spatial and temporal distribution of the $^{222}$Rn exhalation rate from continental surfaces is known, $^{222}$Rn can be used as a quantitative tracer for atmospheric boundary layer mixing, transport model validation (e.g. Jacob and Prather, 1990; Jacob et al., 1997; Taguchi et al., 2011; Williams et al., 2011) and even to estimate regional fluxes of other soil-borne trace gases, by applying the Radon-Tracer-Method (e.g. Levin, 1987; Levin et al., 1999, 2021; Grossi et al., 2018).

Owing to its applicability as atmospheric transport tracer, $^{222}$Rn observations are recommended as supplementary measurement component at all ICOS atmosphere stations (ICOS RI, 2020). For precise atmospheric $^{222}$Rn observations three fundamentally different measurement principles are available: (1) A two-filter dual flow loop detector developed by ANSTO, Australia (Whittlestone and Zahorowski, 1998; Chambers et al., 2011), (2) an electrostatic deposition monitor developed by INTE-UPC, Spain (Grossi et al., 2016), both measuring $^{222}$Rn directly, and (3) static or moving filter detectors, which accumulate $^{222}$Rn progeny on a filter and calculate the atmospheric $^{222}$Rn activity concentration based on assumptions about the secular equilibrium between $^{222}$Rn and its progeny in the atmosphere (e.g. Stockburger and Sittkus, 1966; Polian, 1986; Paatero et al., 1998; Biraud, 2000; Levin et al., 2002; Schmithüsen et al., 2017). This latter $^{222}$Rn measurement principle is applied in the static filter system of the Heidelberg Radon Monitor (HRM, Levin et al., 2002), which measures the atmospheric $^{222}$Rn progeny $^{214}$Po at German ICOS atmosphere stations, and also at sites outside of Germany, including Neumayer station at the Antarctic coast (Weller et al., 2014).

While the HRM static filter radon monitor is a compact and easy to handle instrument, it has some important caveats, which arise from the fact that $^{222}$Rn progeny and not $^{222}$Rn itself are measured. In the atmosphere, newly formed $^{222}$Rn progeny, such as $^{214}$Po, are attached to aerosols and are thus potentially affected by dry or wet deposition processes as well as loss in intake lines if sampling is conducted through long tubing (Levin et al., 2017). Moreover, the state of radioactive equilibrium between progeny and soil-borne $^{222}$Rn in the atmosphere depends on height above ground (Jacobi and Andre, 1963). There may also be some delay in reaching secular radioactive equilibrium of its progeny after rapid increases of the $^{222}$Rn activity concentration. In the past, careful comparison measurements were conducted to evaluate the height-dependence of the disequilibrium (Schmithüsen et al., 2017; Grossi et al., 2020), which turned out to be negligible for measurement heights above ca. 80 m a.g.l. Empirical correction function to compensate for progeny loss in standard 8 mm inner diameter Decabon tubing were determined by extensive laboratory experiments (Levin et al., 2017) and were applied e.g. for the HRM progeny measurements at the German ICOS atmosphere stations to estimate $^{222}$Rn activity concentrations from the measured $^{214}$Po data. A remaining uncertainty is the potential progeny loss due to aerosol scavenging and subsequent wet deposition at very high ambient air humidity which have not been investigated in detail yet.

In this paper we present the atmospheric $^{214}$Po/$^{222}$Rn activity concentration data from German ICOS stations spanning eleven years, from 2014 to 2024, and evaluate these data with respect to potential aerosol scavenging effects, depending on ambient air humidity. Such effects are most frequent at sites where the air intake can reach into clouds, i.e. when a tall tower is located on the top of a mountain or hill. We have therefore developed relative humidity (RH) flagging thresholds for the
60 individual stations, based on re-evaluation of earlier comparison campaign data as well as preliminary regional transport model estimations of the atmospheric $^{222}$Rn activity concentration for the individual sites. The $^{222}$Rn data published along with this paper are flagged according to these thresholds.

## 2 Methods

### 2.1 Radon progeny observations at German ICOS atmosphere stations

Radon progeny observations with the Heidelberg Radon Monitor (HRM) are conducted at eight ICOS atmosphere stations in Germany. All these ICOS atmosphere stations are operated and managed by the Meteorological Observatory Hohenpeißenberg of the German Meteorological Service (DWD) except for the station Schauinsland (SSL) which is run by the German Environment Agency (Umweltbundesamt, UBA). The stations are listed in Tab. 1 together with their coordinates and intake heights. A map of Germany (Fig. 1) shows their almost even distribution across the country. Except for Schauinsland, a mountain station
in the Black Forest (1205 m a.s.l.), where the air intake is about 12 m above ground level (a.g.l.), all other $^{222}$Rn progeny measurements are conducted at towers from intake heights between 93 m and 132 m above ground level. The Torfhaus and Hohenpeißenberg towers are located on top of moderately high hills in the Harz (Torfhaus, 801 m a.s.l.) and in the pre-alpine region of Southern Germany (Hohenpeißenberg, 934 m a.s.l.). All other stations are located in flat terrain at altitudes above sea level (a.s.l.) between 29 m and 110 m. Detailed information about all (German) ICOS stations is available at the ICOS Carbon
Portal (https://www.icos-cp.eu/observations/atmosphere/stations).

### 2.2 Measurement principle and correction for aerosol loss in tubing

The HRM accumulates ambient $^{222}$Rn progeny on a static filter and measures their alpha decay in situ with a surface barrier detector. Half-hourly alpha spectra are integrated and evaluated to separate the $^{214}$Po activity on the filter from other progeny counts, including those of the short-lived radon isotope 220Radon. The methodology and automated algorithm to evaluate the
80 spectra are described in detail by Levin et al. (2002) and Gachkivskyi and Levin (2022). Ambient $^{214}$Po activity concentration is then determined from its activity on the filter by considering the filter-detector geometry, the filter efficiency and the mass flow of air through the filter, using the so-called Stockburger algorithm (Stockburger, 1960; Stockburger and Sittkus, 1966). Subsequently, a correction of progeny loss in the intake tubing (Decabon, 8 mm inner diameter), which can be up to 158 m long (Tab. 1), is then applied following Levin et al. (2017). The length-dependent loss of progeny is approximated by an exponential
function





$$R(L) = \exp^{-\frac{L}{L_0}} \tag{1}$$

with L being the length of the tubing and $L_0$ a constant that was experimentally determined in the laboratory. For a given length of tubing, progeny loss was further observed to be dependent on ambient activity concentration. This could be approximated with a saturation curve as follows:

$$\frac{c_{measured}}{c_{ambient}} = R(L) - A \cdot \exp^{-\frac{c_{measured}}{c_0}} \tag{2}$$

where the experimentally determined empirical parameters $A$ and $c_0$ are constants, and $R(L)$ is calculated from Eq. 1 for line length L. The constants in Eqs. 1 and 2 have been reported by Levin et al. (2017) to $L_0 = 415$ m, $A = 0.41$ and $c_0 = 0.92$ Bq m$^{-3}$. The line loss correction was applied to all stations with air intake lines longer than 15 meters, which represents the shortest tested line length for aerosol loss (Levin et al., 2017). No aerosol loss due to the tubing is assumed for lines shorter than 15 meters.

## 2.3 Calibration and $^{214}$Po/$^{222}$Rn disequilibrium

In a theoretical evaluation, Jacobi and Andre (1963) estimated the height-dependence of the disequilibrium between $^{214}$Po and $^{222}$Rn. This disequilibrium depends on atmospheric stability and decreases with height, approaching equilibrium at levels of about 100 m a.g.l. In an international comparison experiment of the HRM measuring $^{214}$Po with direct $^{222}$Rn measurement using an ANSTO monitor, Schmithüsen et al. (2017) showed that usually equilibrium between $^{222}$Rn and $^{214}$Po measurements in ambient air is reached at intake heights exceeding 60 m above ground level. However, they found a calibration difference between the two systems of about 10%. A similar result with a slightly smaller difference between monitors was obtained in a comparison campaign conducted in 2017 at the ICOS station Saclay, France, where parallel measurements at the 100 m intake height were conducted (Grossi et al., 2020). As there is currently no official standard for atmospheric $^{222}$Rn measurements, we did not apply any scaling factors to our HRM measurements e.g. to adjust them to the ANSTO scale. Instead, we run each HRM in parallel to our routine monitor in Heidelberg for a few weeks and determine an internal calibration factor that is applied to have comparable measurements within the HRM network. We also did not correct for disequilibrium any of the HRM data from ICOS tower stations with intake heights > 90 m a.g.l., assuming that the atmospheric $^{222}$Rn activity concentration is equal to that of the measured $^{214}$Po activity concentration. For the Schauinsland data, we applied a factor of 1/0.8 to the $^{214}$Po measurements to estimate ambient $^{222}$Rn activity concentrations. This factor corresponds to that reported by Schmithüsen et al. (2017, Table 2).

For the Saclay 2017 comparison campaign, Grossi et al. (2020) showed that the ratio of HRM-measured $^{214}$Po and $^{222}$Rn measured with the ARMON monitor from Universitat Politècnica de Catalunya (UPC) decreased at relative humidity (RH) values close to 100%. We have re-evaluated the data from this campaign, also correcting the $^{222}$Rn measurements from the ANSTO detector for standard temperature and pressure (STP, T = 293.15 K, P = 1000 hPa), to make them comparable with





the HRM measurements. These re-evaluated HRM and ANSTO data are displayed in Fig. 2, panel (a) together with the RH measured during that period. The relation between HRM $^{214}$Po and ANSTO $^{222}$Rn data is displayed in the panel (c). Here we selected the HRM data by distinguishing all hourly values by using co-measured RH data. The pink dots in the panel (c) with RH > 98% clearly deviate from the regression line, with the values from the HRM being lower than the data from ANSTO.

This would be consistent with the $^{214}$Po aerosols being scavenged during the high RH events. The panel (b) shows the Pearson correlation coefficient between the two data sets dependent on the data selection by RH threshold. This means, that data measured at a lower RH than the threshold is included in the fit calculation. We use the correlation coefficient as the relevant parameter to be investigated, because there may be other reasons for discrepancies in absolute concentrations between the two measurement systems, such as uncertainties in calibration or instrumental issues that cause systematic biases. A correlation

provides a useful tool for examining the relative differences between datasets. A small decrease of the correlation coefficient is observed from 90% to 99% RH, followed by a steep decrease towards 100% RH. Therefore, for this comparison campaign, a relative humidity threshold of 98% would be set to determine a potential "calibration factor" between the two instruments is to be determined, accounting for uncertainty of RH data and ensure the compatibility of the data. Excluding data measured at RH > 98%, we obtain an excellent agreement of almost 1:1 (0.97±0.01) with no significant offset between the HRM and the

ANSTO measurements.

## 2.4 Humidity threshold for the German $^{214}$Po/$^{222}$Rn network

No direct $^{222}$Rn observations are available to determine a relative humidity threshold for the German stations with HRM progeny measurements, as was done for the Saclay 2017 campaign (Section 2.3). Therefore, we used a threshold based on the comparison with atmospheric transport model calculations of $^{222}$Rn. We used the Stochastic Time-Inverted Lagrangian

Transport model STILT (Lin et al., 2003) that was implemented at the ICOS Carbon Portal (https://www.icos-cp.eu/about-stilt) to calculate footprints and atmospheric $^{222}$Rn activity concentrations for all ICOS stations. The radon flux estimates for Europe of Karstens and Levin (2023), providing daily mean $^{222}$Rn fluxes for the years 2017-2023, were used to calculate hourly $^{222}$Rn activity concentrations. We then calculated month-wise correlation coefficients between observed $^{214}$Po and modelled $^{222}$Rn activity concentrations by successively adding data at increasing observed RH (Kohler et al., 2018; Kubistin et al.,

2024a, b, c, d, e, f, g; Schmidt et al., 2024; UBA, 2024), similar to the method used for the Saclay 2017 comparison campaign data (Fig. 2 panels (b) and (d)). Note that this comparison between measured $^{214}$Po and simulated $^{222}$Rn activity concentrations was conducted only to diagnose thresholds for potential $^{214}$Po loss at the individual stations. Quantitative evaluations, e.g. of the absolute agreement between model simulations and observations were not target of this comparison, as we are aware that the $^{222}$Rn flux map as well as transport in the STILT model have significant uncertainties. We also do not expect good

correlations and as smooth curves as were obtained from the comparison of the two measured records at Saclay (Fig. 2, panel (b)). Nevertheless, we assume that a significant and abrupt decrease of correlation coefficient at high RH should also become visible as a consequence of loss of $^{214}$Po. This could then be used to pinpoint a threshold of maximum RH at which the radioactive equilibrium of $^{222}$Rn and its progeny $^{214}$Po is still valid.





Figure 3 shows typical examples of correlation coefficients between modelled $^{222}$Rn and measured $^{214}$Po (panels (a) and
150 (e)) together with RH distributions at three out of eight stations in the German ICOS network for those hourly situations where
$^{214}$Po measurements are available. Note that for the mountain stations Schauinsland (SSL, panels (d) and (h)), Torfhaus and
Hohenpeißenberg (resp. TOH and HPB, Figs. A1-A12 in Supplementary Material), we did not use STILT results for the actual
tower heights (see Tab. 1), but for higher elevations of 300 m (TOH, HPB) and 230 m (SSL), as for these virtual heights model
results show better agreement with the observations (Geels et al., 2007; Oney et al., 2015; Kountouris et al., 2018). Panels
(b) – (d) in figure 3 show the data from April 2024, a month with a relatively high frequency of RH > 95% at the mountain
station of SSL and very few RH values above 90% at the tower stations in flat terrain. The majority of the RH measurements
in this month are centered around 60% RH or even lower. All stations show a correlation between measured and modelled data
higher than 0.6 if enough data is provided. This could be explained by the fact that for the relatively dry conditions the $^{214}$Po
concentrations represent the $^{222}$Rn values quite well. But it can also be clearly seen that SSL the correlation coefficient steeply
decreases from about 0.65 to about 0.51 if values with RH > 98% are included in the calculation and if the number of these
high RH situations becomes significant. The KIT and LIN stations remain largely unchanged in this regard.

In the panels (f) – (h) of figure 3 the corresponding data from November 2022 are shown. In this month RH close to saturation
is observed at all stations. However, it is only at very high RH >98% that the correlation coefficient decreases also at the tower
stations in flat terrain. Overall, there is a tendency towards higher correlation coefficients between model and observations at
165 the stations located in flat terrain (GAT, STE, KIT, JUE and LIN) compared to the mountain sites (TOH, SSL, HPB). This is
because trace gas concentrations at mountain stations, due to their complex topography, are generally less well represented in
the model than those in flat terrain. A similar behaviour is observed for the other stations of the German ICOS network and
other months of 2022, where we have good overlap of measurements and STILT model results (Figs. A1-A12 in Supplementary
Material). In some rare cases, where most data were measured at RH > 98%, a sharp increase in correlation can be observed
for high RH values for some stations (e.g. December 2022, Fig. A12). Based on these findings, we suggest to only use $^{214}$Po
data for the calculation of $^{222}$Rn activity concentrations measured at ambient RH < 98%. Secular equilibrium between $^{214}$Po
and $^{222}$Rn can only be assumed for measurements made at RH lower than this threshold. It should be noted that the collection
of data published together with this manuscript (Deutscher Wetterdienst et al., 2024a, b, c, d, e, f, g, h) contain all valid $^{214}$Po
measurements, and we provide a value for $^{222}$Rn only for conditions, where full equilibrium between $^{214}$Po and $^{222}$Rn based
on the measured RH can be assumed.

## 3  Results

Daily mean $^{214}$Po data from all stations are shown in the left panel of Fig. 4, while the data remaining after relative humidity
flagging (where available) according to the procedure described in Sec. 2.4 are shown in the right panel of Fig. 4. Only mean
values for days where more than 70% of the hourly data are available or remained after flagging are displayed. Where no RH
data is available, the $^{214}$Po data has been flagged. At some of the sites there are large gaps in the $^{214}$Po data, which are due to
initial handling and logistical problems or occasional HRM failures. After flagging, data gaps become more frequent, especially





during winter at high RH and at mountain stations like SSL and TOH, where the high RH events are more prevalent in general. On average, we observe the highest $^{214}$Po (and $^{222}$Rn) activity concentrations at KIT located in the upper Rhine valley. Even the mountain stations, which usually have longer air mass transport from the $^{222}$Rn source and hence lower concentrations,

SSL and HPB show elevated concentrations compared to the rest of the network, with both of them and KIT located in the south of Germany. The lowest activity concentrations are found at GAT, STE and TOH with JUE showing slightly higher $^{214}$Po activities. All these stations are situated in the northern half of Germany. This south-north gradient of the $^{222}$Rn flux is consistent with the radon flux estimates for Europe derived from uranium soil content, soil properties and moisture content by Karstens and Levin (2023). The only exception to this longitudinal scheme is the LIN station that, however, has a region of

elevated radon soil emission values to the south of the station.

At all stations we find a clear seasonal cycle with highest values in summer and autumn and lowest activity concentrations in winter and spring (Fig. 5 shows KIT, SSL and LIN, and A13 in Supplementary Material depicts all stations). This seasonality is mainly caused by the seasonality of the radon exhalation rate from soils, which, on the seasonal time scale, is mainly driven by soil moisture (Karstens et al., 2015). The high $^{222}$Rn values in summer and autumn are also due to stable atmospheric

conditions in these seasons during nights (cf. Fig. 6) as well as to more frequent high-pressure systems bringing continental air masses to Germany.

The mean diurnal cycles of $^{222}$Rn from January to December 2022 are displayed in Fig. 6. Only in summer and autumn (i.e. from April to September) we observe on average significant diurnal variations at the tower stations in flat terrain with maximum values early in the morning after sunrise, i.e. shortly before the atmosphere becomes well-mixed and $^{222}$Rn that

was accumulated close to the ground is mixed into higher levels of the troposphere. At HPB the morning maximum occurs with a few hours delay, as ground level radon is not only transported through vertical convection but also by upslope winds from the surrounding valleys to the sampling site. A similar diurnal variation is observed at SSL in some of the summer and autumn months. This behaviour of $^{222}$Rn had already been observed earlier by Levin (1987). During winter and early spring, systematic diurnal variations are missing at all sites as the variations of the $^{222}$Rn activity concentration are much more

dominated by synoptic variations.

## 4 Discussion and Conclusions

Currently, there are eight ICOS stations within Germany measuring atmospheric $^{214}$Po/$^{222}$Rn activity concentrations; these are fairly even distributed over the country with 5 tall towers in flat terrain and 3 stations on mountain tops (two of which are sampling air from more than 90 m above ground level). Observations are conducted with the static filter Heidelberg Radon

Monitor, measuring atmospheric $^{222}$Rn progeny. At all tower stations sampling air from intake heights more than 90 m above ground level, progeny can be assumed in equilibrium with atmospheric $^{222}$Rn. Only at very high relative humidity > 98% this is not necessarily the case, which we were able to demonstrate by comparison with simulated $^{222}$Rn activity concentrations. We, therefore, recommend excluding the data measured at very high RH in any applications where $^{222}$Rn shall be used as a tracer for boundary layer transport e.g. for atmospheric transport model validation or for the application of the Radon-Tracer-Method





(RTM Levin et al., 2021). In this latter application, the missing data at high RH will probably not cause additional uncertainty in RTM-based flux estimates, because these very humid atmospheric situations with marine air reaching the stations normally show little correlation between GHGs concentrations and $^{222}$Rn, and such situations would have been discarded. In the case of atmospheric transport model validation, however, it is essential to use appropriately flagged radon data, where the equilibrium between radon and its progenies can be ensured, even if this results in data gaps. The scavenging of aerosols in small droplets,

followed by wet deposition and rain out, can cause a significant reduction in radon progenies, which leads to discrepancies with radon signal. These findings further highlight the limitations of the HRMs and need for direct $^{222}$Rn measurements at the high altitude sites where clouds frequently coincide with the intake heights. The amount of flagged data ranges from 3% to 55% in the wetter winter months (December – February, average 24%), with highest flagged percentage in TOH (51%) and SSL (55%), and lowest in KIT (9%) and JUE (3%). During the dryer summer months (June – August), percentage of flagged

observations is significantly lower (average 7%), with the same stations for highest and lowest amounts (high: TOH – 13%, SSL – 16%, low: KIT – 1%, JUE – 1%).

In view of the rather small percentage of the data where the $^{214}$Po data measured with the HRM were most probably not in secular equilibrium with $^{222}$Rn (with exception of SSL and TOH stations during winter months), and since the absolute agreement between the HRM and other instruments measuring $^{222}$Rn directly was in the order of ±10% during intercomparison

campaigns, we conclude that our method is suitable for sampling heights above 80-90 m a.g.l for the stations in the flat terrain. The situation may be different for lower intake heights, where we cannot generally assume radioactive equilibrium between $^{222}$Rn and its progenies. For example, in the case of our station in Heidelberg, where air is collected from the roof of the building at 30 m a.g.l., we have to apply a mean correction for disequilibrium, multiplying the $^{214}$Po activity concentration with a corresponding factor that was based on comparison measurements with an ANSTO detector (i.e. 1.11 Schmithüsen

et al., 2017; Levin et al., 2021). A correction for disequilibrium also must be applied to the Schauinsland $^{214}$Po data based on comparison with direct $^{222}$Rn measurements (Xia et al., 2010). As the disequilibrium between $^{214}$Po and $^{222}$Rn depends on atmospheric stability (Jacobi and Andre, 1963; Cuntz, 1997), applying just one mean factor adds uncertainty to the $^{222}$Rn data. Measuring $^{222}$Rn gradients below 80-90 m a.g.l. is, therefore, not recommended with the HRM, unless a careful comparison with an instrument measuring $^{222}$Rn directly has been conducted for each height level, in order to apply a disequilibrium

correction based on direct observations.

## 5 Data availability

The $^{222}$Rn observations are available at the ICOS Carbon Portal (https://doi.org/10.18160/Q2M8-B1HJ, Deutscher Wetterdienst et al., 2024a, b, c, d, e, f, g, h).



**Appendix A**

**A1**

*Author contributions.*  MG evaluated all raw data, conducted the relative humidity selection and prepared the figures. UK made the STILT model runs. BF, ML, DK, JMW provided the observational data. IL together with UK and MG designed the study and prepared the manuscript. All authors discussed and reviewed the manuscript.

*Competing interests.*  The authors declare that they have no conflict of interest.

*Acknowledgements.*  We wish to thank the German Meteorological Service station technicians for their support in radon measurement and maintenance. The HRM instrumentation was partly funded by the Federal Ministry of Education and Research and the Federal Ministry for Digital and Transport. This study was supported by the ICOS Central Radiocarbon Laboratory, which is funded by the German Federal Ministry for Digital and Transport. Ute Karstens received funding from the project 19ENV01 traceRadon, part of the EMPIR program that is co-financed by the Participating States and from the European Union's Horizon 2020 research and innovation program. We would like to

express our sincere gratitude to Samuel Hammer for his valuable time and insightful comments on this manuscript.



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



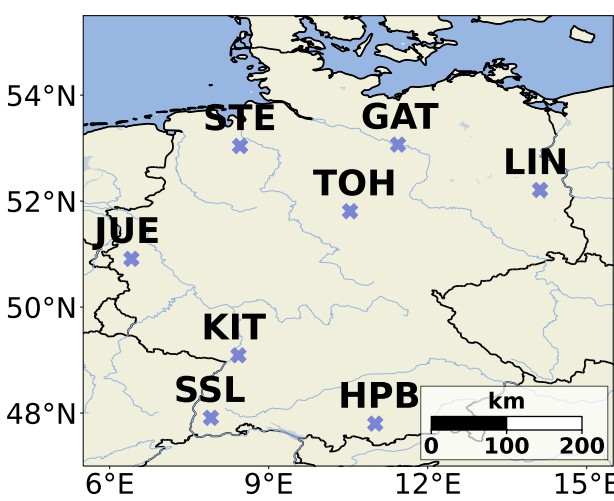

**Figure 1.** Radon measurements in Germany are conducted with the HRM at eight ICOS atmosphere stations. All stations except for SSL are tall towers operated by DWD (Radon intake heights between 93 m and 132 m a.g.l.). SSL has its air intake at 12 m a.g.l. and is operated by the German Environment Agency.

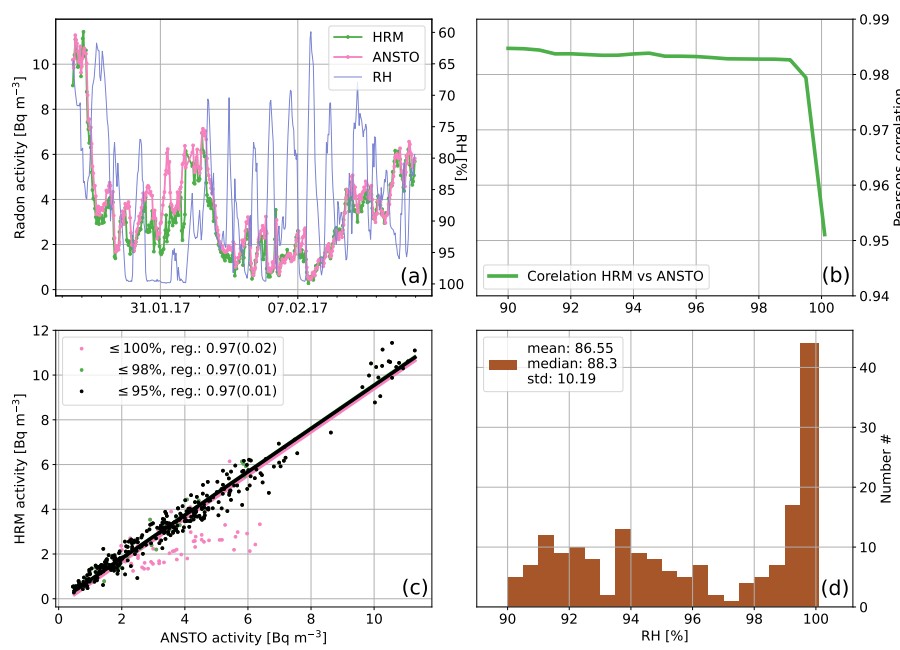

**Figure 2.** Activity concentration measured with the ANSTO monitor and with the HRM during the 2017 comparison campaign at Saclay tower at the 100m level. The panel (a) shows the original records together with relative humidity (the frequency distribution of RH is shown in the panel (d)). The panel (c) displays the regression lines between the two datasets with some data being excluded depending on the co-measured RH values. The data is obtained at RH less than or equal to 95% and the additions to this set with RH less than or equal to 98% are highlighted in green. Data measured at RH > 98% up to 100% are marked in pink. The data with RH > 98% (in pink) clearly fall off the regression line. The panel (b) shows the development of the Pearson correlation coefficient when successively more data measured at higher RH are included in the correlation calculation.

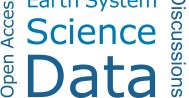

**Figure 3.** The panels (a) and (e) show the development of the correlation coefficient between $^{214}$Po observations and simulated $^{222}$Rn data at three German ICOS stations (KIT, LIN, SSL) for April and for November 2022 respectively if data with increasing RH are successively included in the calculation of the correlation. The panels (b) – (d) and (f) – (h) show the RH distributions for April and November respectively at these stations that were included in the calculations. The arrows with numbers indicate values that exceed the y-axis range (usually for high RH). For each station the correlation coefficient was calculated only if the number of filtered data points was higher than 10% of all values for the given month to avoid the stochastic behavior of the curves in the panels (a) and (e) associated with small data sets.



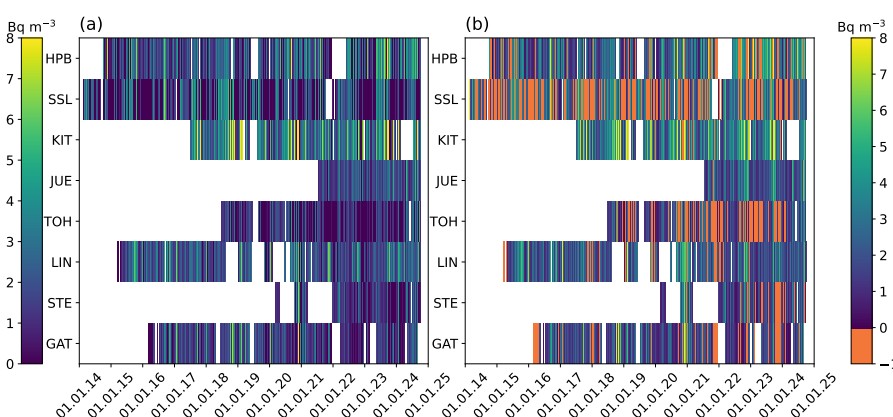

**Figure 4.** Panel (a) shows daily mean atmospheric $^{214}$Po activity concentrations in Bq m-3 at German ICOS stations. The daily values are plotted only if more than 70% of the hourly data was available. The panel (b) shows all valid data where measurements were conducted at RH below the respective thresholds and thus represent the best estimates of the RH-selected $^{222}$Rn data. The periods where less than 70% of the hourly data remained after flagging are shown in orange (with values of -1).



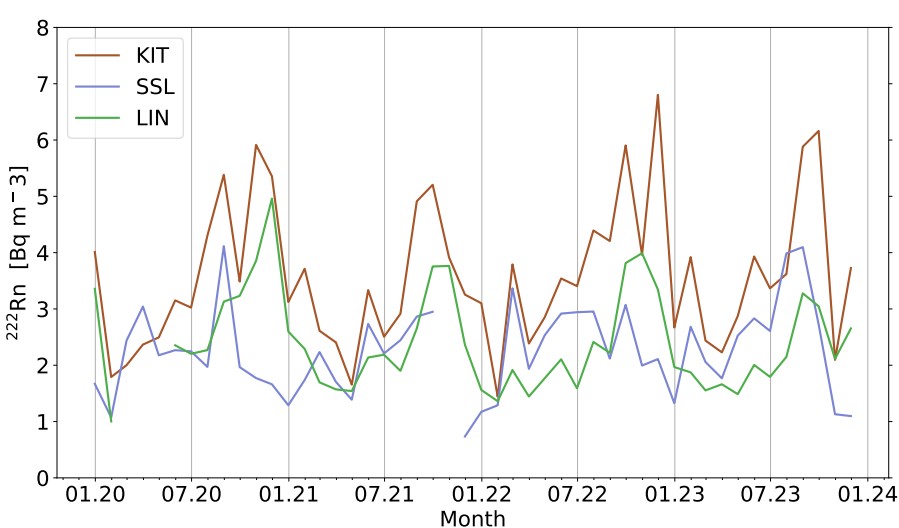

**Figure 5.** Seasonal cycle of $^{222}$Rn monthly means at KIT, SSL and LIN in the time period of 2020-2023. Note that monthly values are only shown if more than 70% of the daily mean values were available for the respective month.


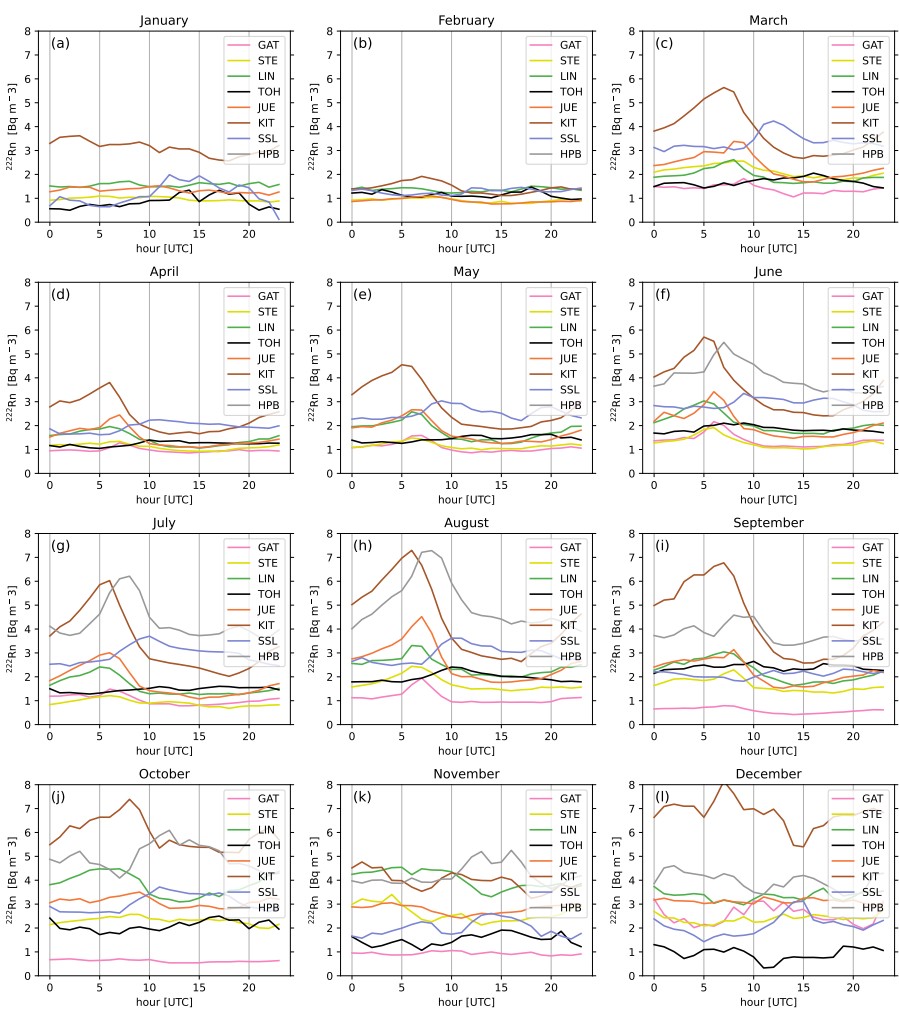

**Figure 6.** Mean diurnal cycles of $^{214}$Po-based $^{222}$Rn for all months in 2022 where data is available.

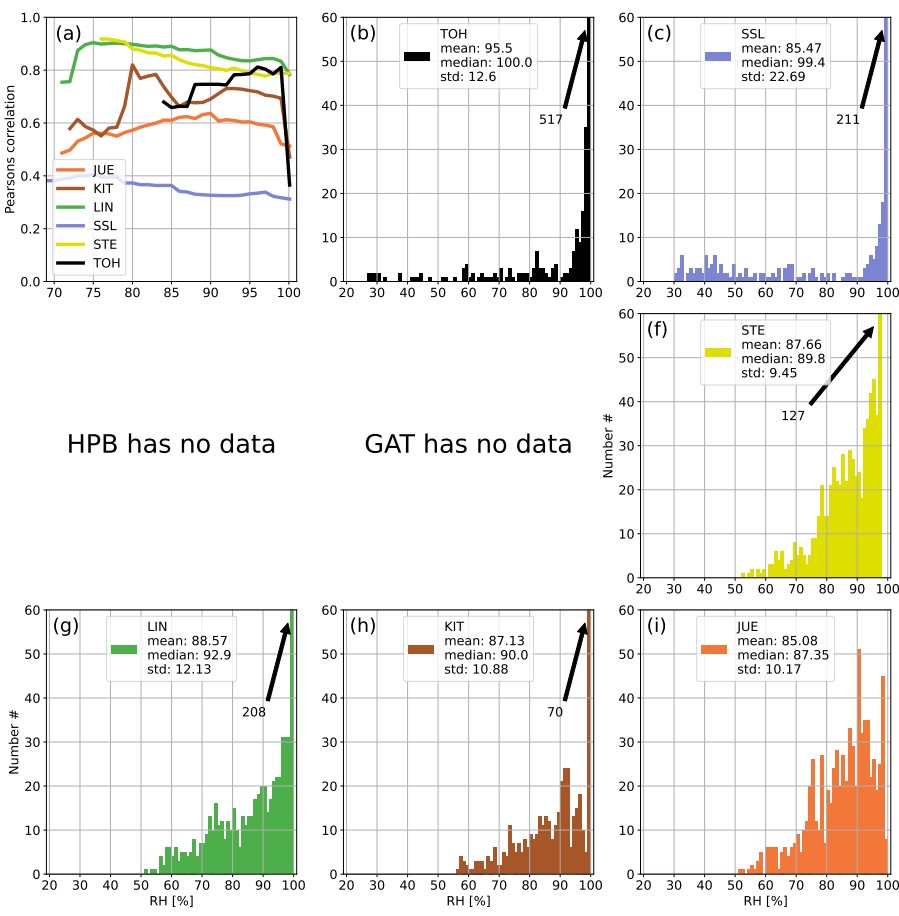

**Figure A1.** The panel (a) shows the development of the correlation coefficient between $^{214}$Po observations and simulated $^{222}$Rn data at all German ICOS stations if data with increasing RH are successively included in the calculation of the correlation. The panels (b) – (i) show the distributions of hourly RH values at the stations of the network. The arrows with numbers indicate values that exceed the y-axis range (usually for high RH). For each station the correlation coefficient was calculated only if the number of filtered data points was higher than 10% of all values for the given month to avoid the stochastic behaviour of the curves in the first subplot associated with small data sets. January 2022.

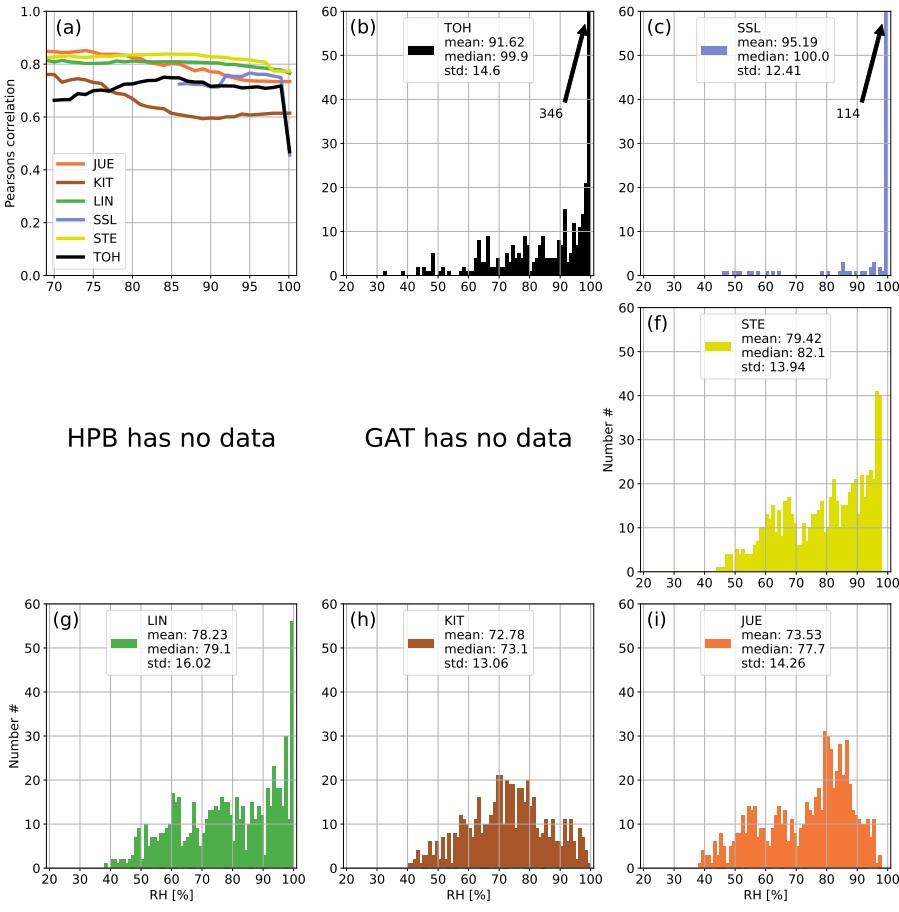

**Figure A2.** The panel (a) shows the development of the correlation coefficient between $^{214}$Po observations and simulated $^{222}$Rn data at all German ICOS stations if data with increasing RH are successively included in the calculation of the correlation. The panels (b) – (i) show the distributions of hourly RH values at the stations of the network. The arrows with numbers indicate values that exceed the y-axis range (usually for high RH). For each station the correlation coefficient was calculated only if the number of filtered data points was higher than 10% of all values for the given month to avoid the stochastic behaviour of the curves in the first subplot associated with small data sets. February 2022.

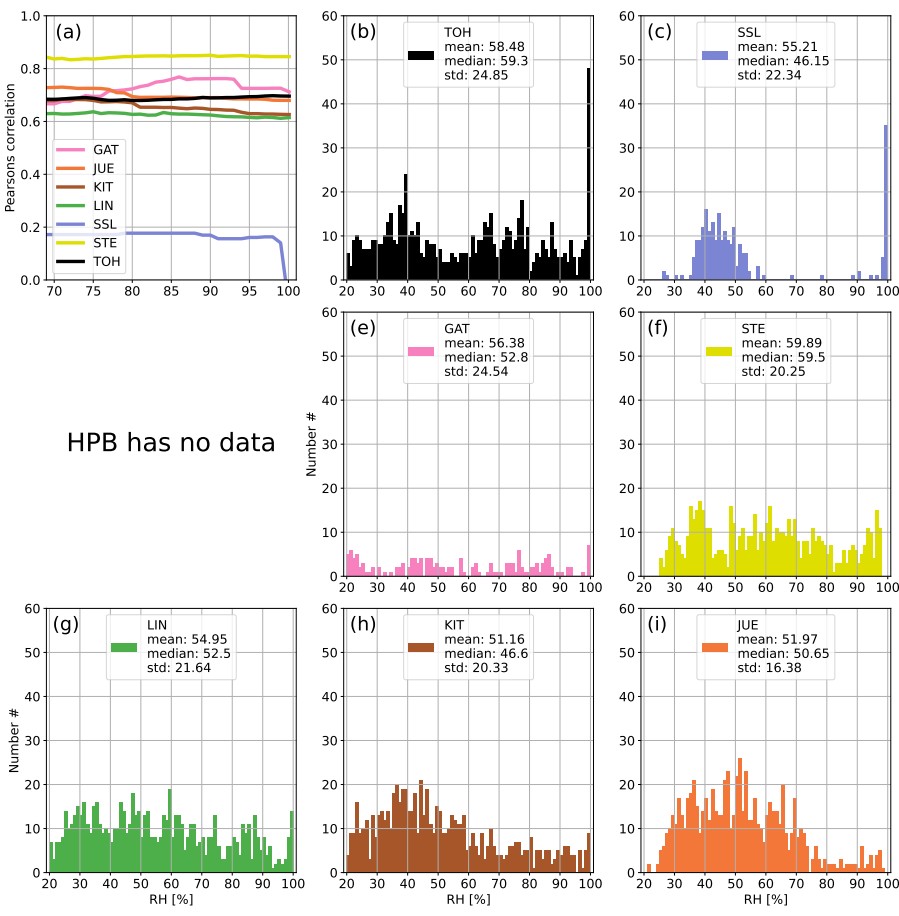

**Figure A3.** The panel (a) shows the development of the correlation coefficient between $^{214}$Po observations and simulated $^{222}$Rn data at all German ICOS stations if data with increasing RH are successively included in the calculation of the correlation. The panels (b) – (i) show the distributions of hourly RH values at the stations of the network. The arrows with numbers indicate values that exceed the y-axis range (usually for high RH). For each station the correlation coefficient was calculated only if the number of filtered data points was higher than 10% of all values for the given month to avoid the stochastic behaviour of the curves in the first subplot associated with small data sets. March 2022.

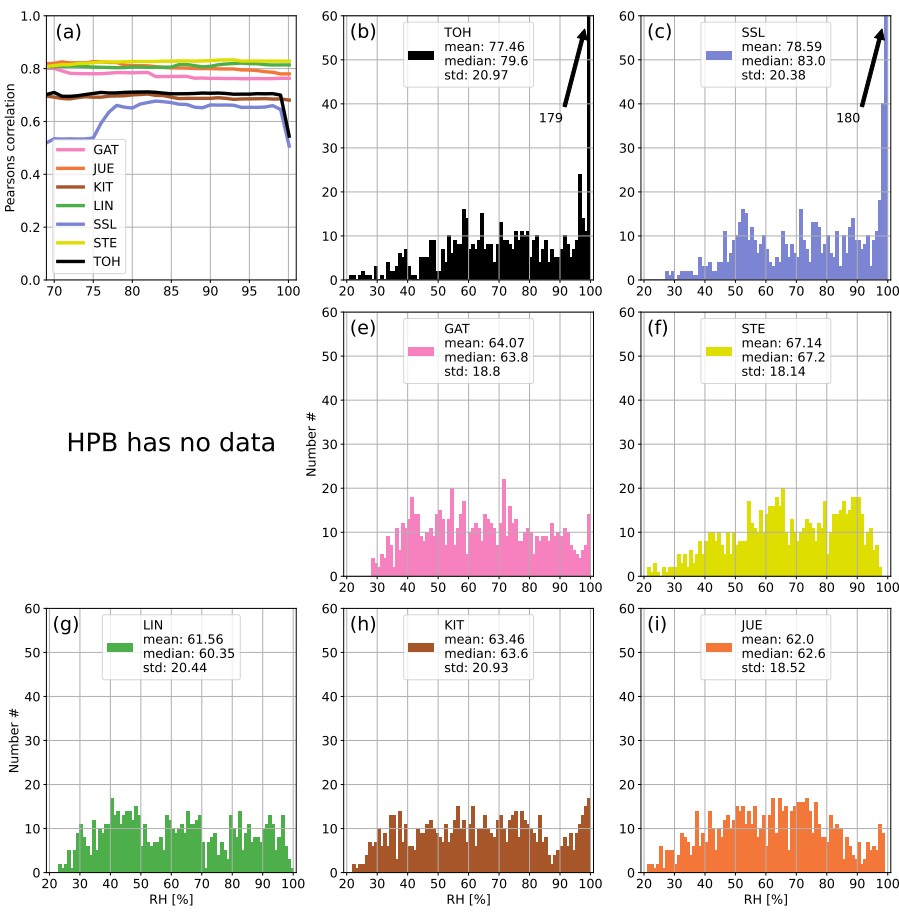

**Figure A4.** The panel (a) shows the development of the correlation coefficient between $^{214}$Po observations and simulated $^{222}$Rn data at all German ICOS stations if data with increasing RH are successively included in the calculation of the correlation. The panels (b) – (i) show the distributions of hourly RH values at the stations of the network. The arrows with numbers indicate values that exceed the y-axis range (usually for high RH). For each station the correlation coefficient was calculated only if the number of filtered data points was higher than 10% of all values for the given month to avoid the stochastic behaviour of the curves in the first subplot associated with small data sets. April 2022.

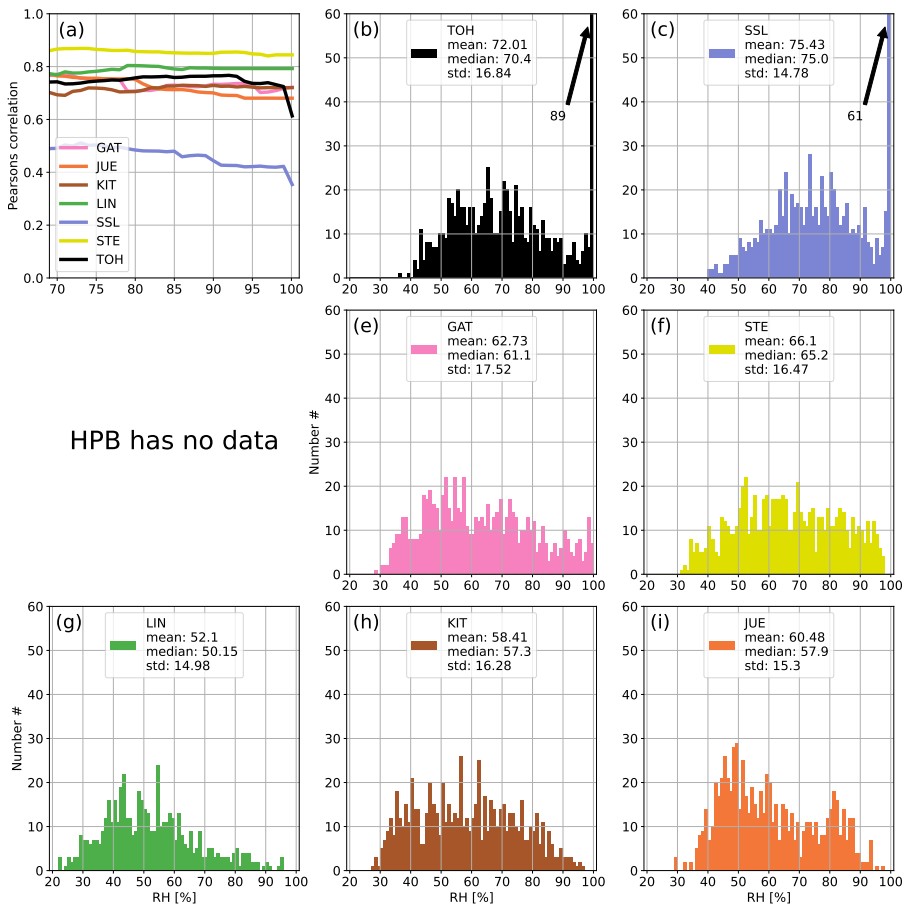

**Figure A5.** The panel (a) shows the development of the correlation coefficient between $^{214}$Po observations and simulated $^{222}$Rn data at all German ICOS stations if data with increasing RH are successively included in the calculation of the correlation. The panels (b) – (i) show the distributions of hourly RH values at the stations of the network. The arrows with numbers indicate values that exceed the y-axis range (usually for high RH). For each station the correlation coefficient was calculated only if the number of filtered data points was higher than 10% of all values for the given month to avoid the stochastic behaviour of the curves in the first subplot associated with small data sets. May 2022.

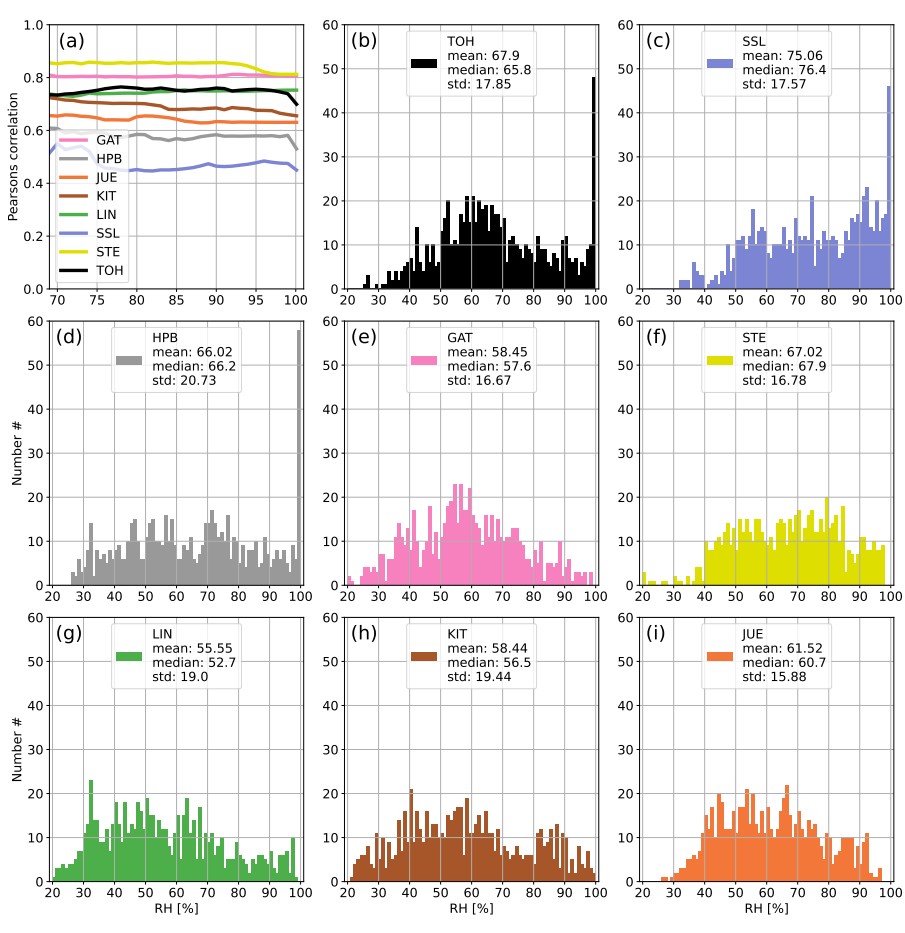

**Figure A6.** The panel (a) shows the development of the correlation coefficient between $^{214}$Po observations and simulated $^{222}$Rn data at all German ICOS stations if data with increasing RH are successively included in the calculation of the correlation. The panels (b) – (i) show the distributions of hourly RH values at the stations of the network. The arrows with numbers indicate values that exceed the y-axis range (usually for high RH). For each station the correlation coefficient was calculated only if the number of filtered data points was higher than 10% of all values for the given month to avoid the stochastic behaviour of the curves in the first subplot associated with small data sets. June 2022.

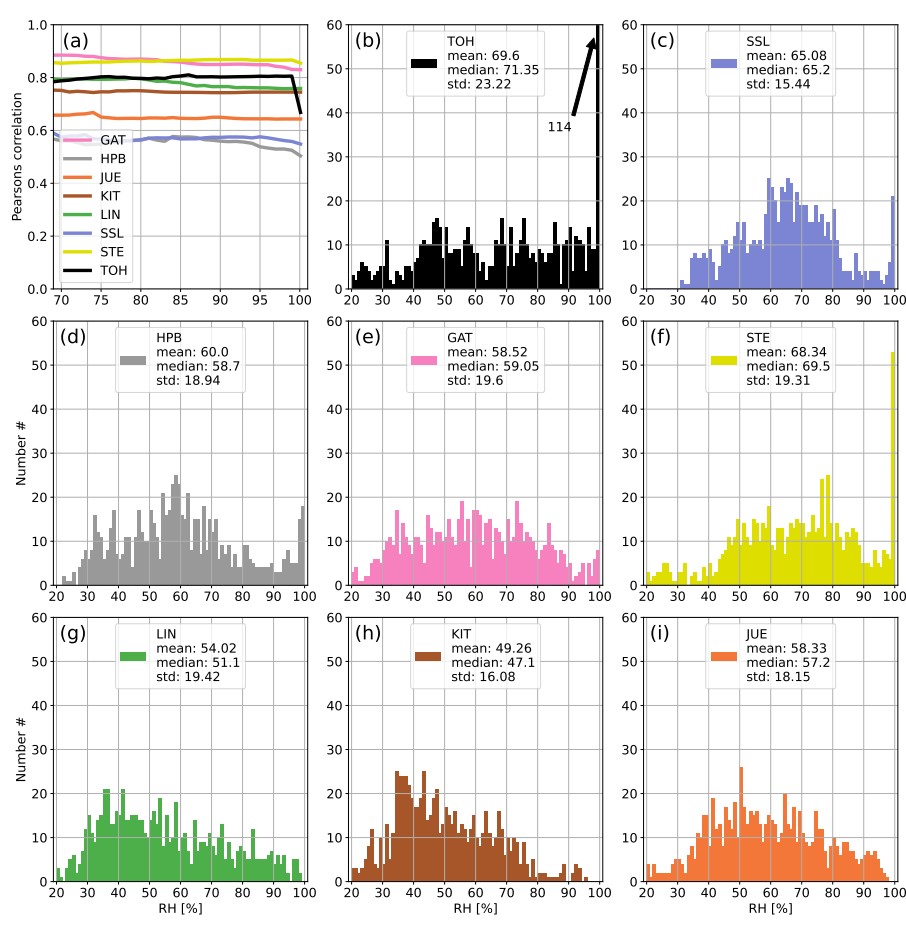

**Figure A7.** The panel (a) shows the development of the correlation coefficient between $^{214}$Po observations and simulated $^{222}$Rn data at all German ICOS stations if data with increasing RH are successively included in the calculation of the correlation. The panels (b) – (i) show the distributions of hourly RH values at the stations of the network. The arrows with numbers indicate values that exceed the y-axis range (usually for high RH). For each station the correlation coefficient was calculated only if the number of filtered data points was higher than 10% of all values for the given month to avoid the stochastic behaviour of the curves in the first subplot associated with small data sets. July 2022.

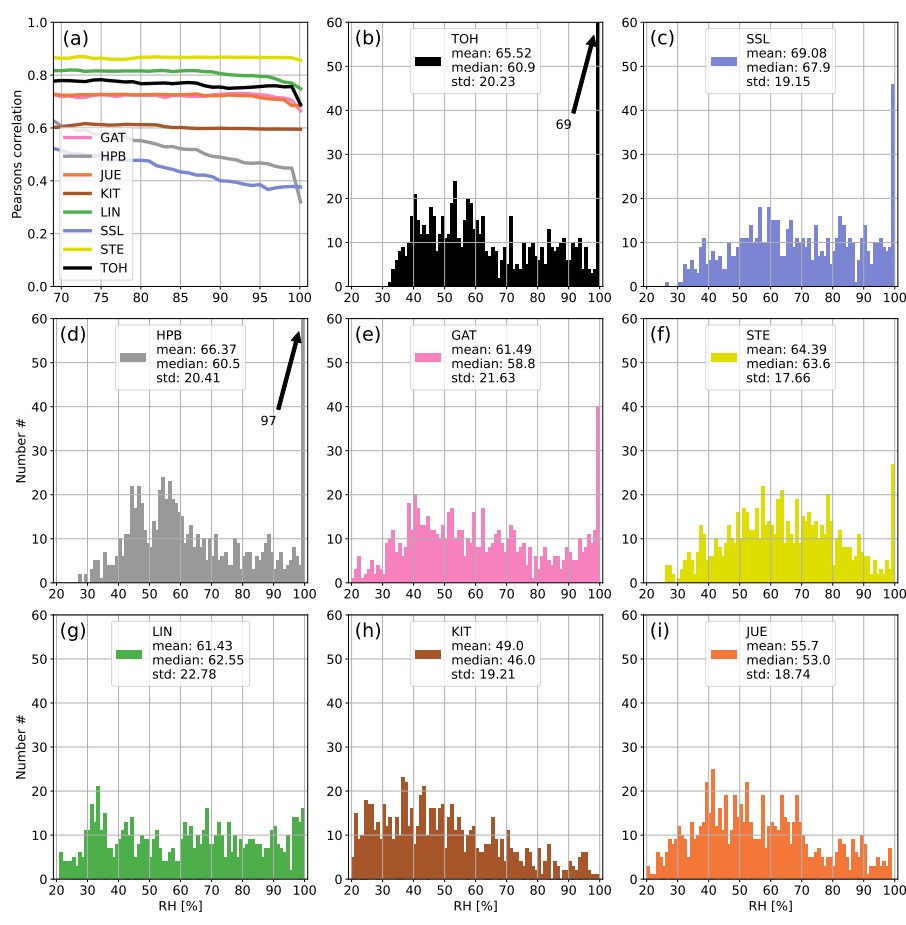

**Figure A8.** The panel (a) shows the development of the correlation coefficient between $^{214}$Po observations and simulated $^{222}$Rn data at all German ICOS stations if data with increasing RH are successively included in the calculation of the correlation. The panels (b) – (i) show the distributions of hourly RH values at the stations of the network. The arrows with numbers indicate values that exceed the y-axis range (usually for high RH). For each station the correlation coefficient was calculated only if the number of filtered data points was higher than 10% of all values for the given month to avoid the stochastic behaviour of the curves in the first subplot associated with small data sets. August 2022.

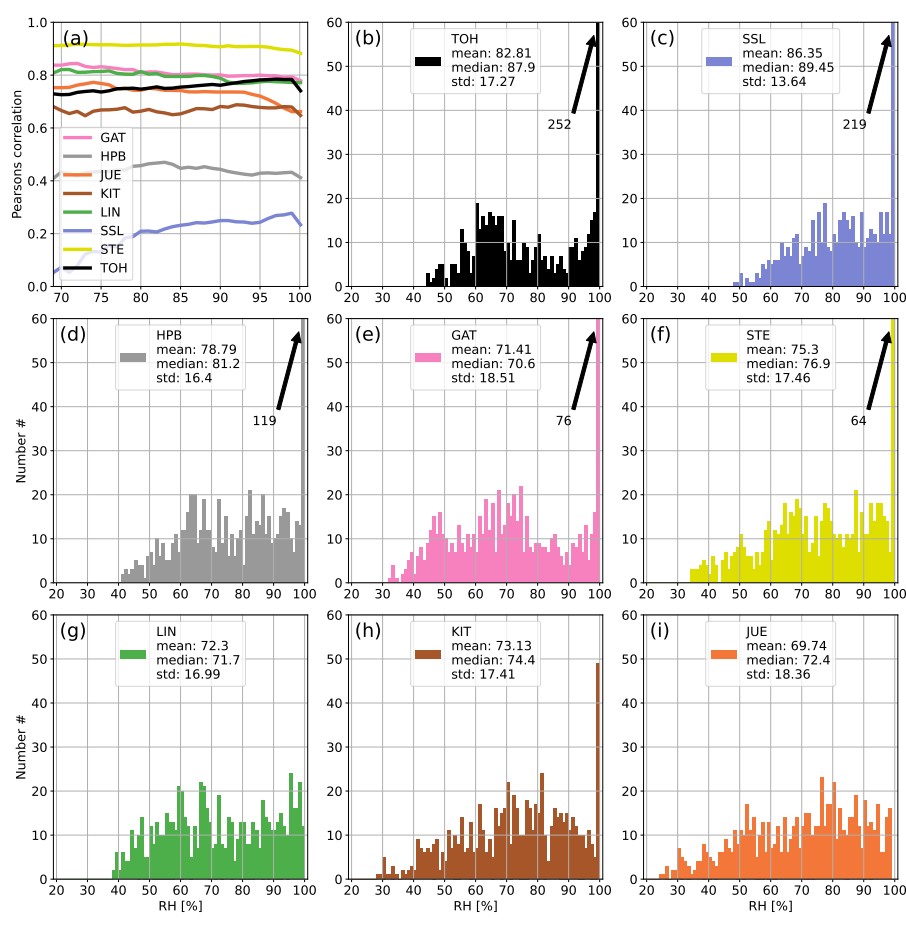

**Figure A9.** The panel (a) shows the development of the correlation coefficient between $^{214}$Po observations and simulated $^{222}$Rn data at all German ICOS stations if data with increasing RH are successively included in the calculation of the correlation. The panels (b) – (i) show the distributions of hourly RH values at the stations of the network. The arrows with numbers indicate values that exceed the y-axis range (usually for high RH). For each station the correlation coefficient was calculated only if the number of filtered data points was higher than 10% of all values for the given month to avoid the stochastic behaviour of the curves in the first subplot associated with small data sets. September 2022.

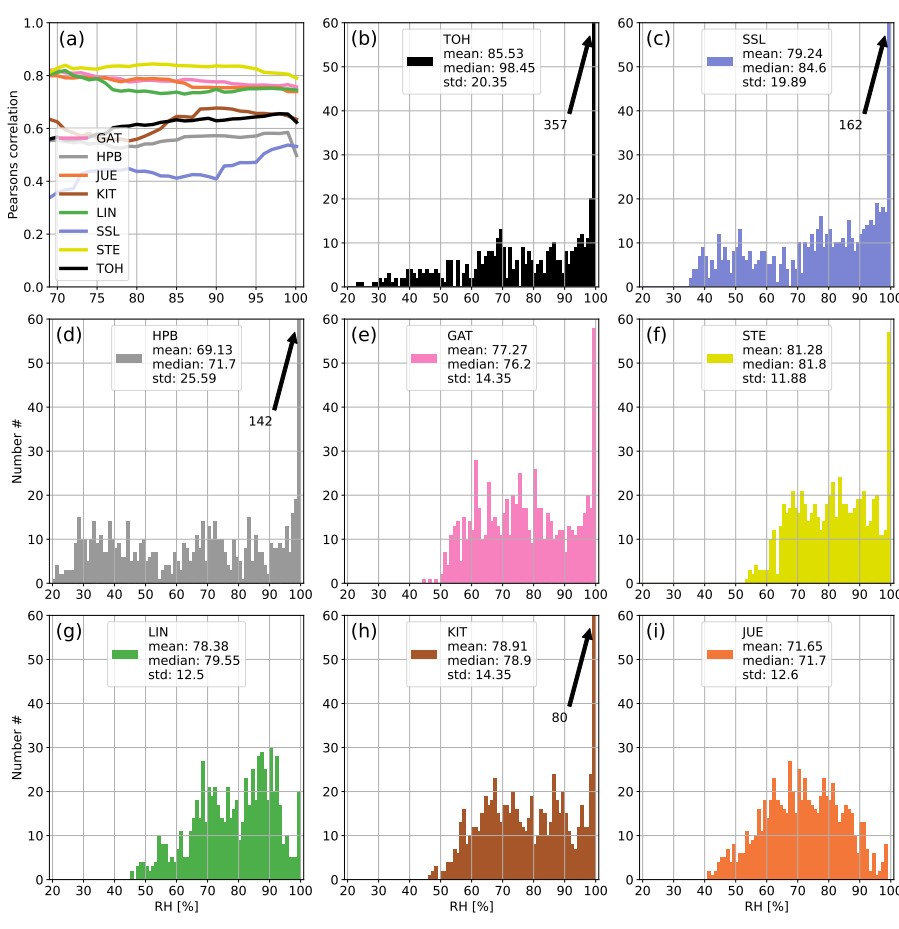

**Figure A10.** The panel (a) shows the development of the correlation coefficient between $^{214}$Po observations and simulated $^{222}$Rn data at all German ICOS stations if data with increasing RH are successively included in the calculation of the correlation. The panels (b) – (i) show the distributions of hourly RH values at the stations of the network. The arrows with numbers indicate values that exceed the y-axis range (usually for high RH). For each station the correlation coefficient was calculated only if the number of filtered data points was higher than 10% of all values for the given month to avoid the stochastic behaviour of the curves in the first subplot associated with small data sets. October 2022.

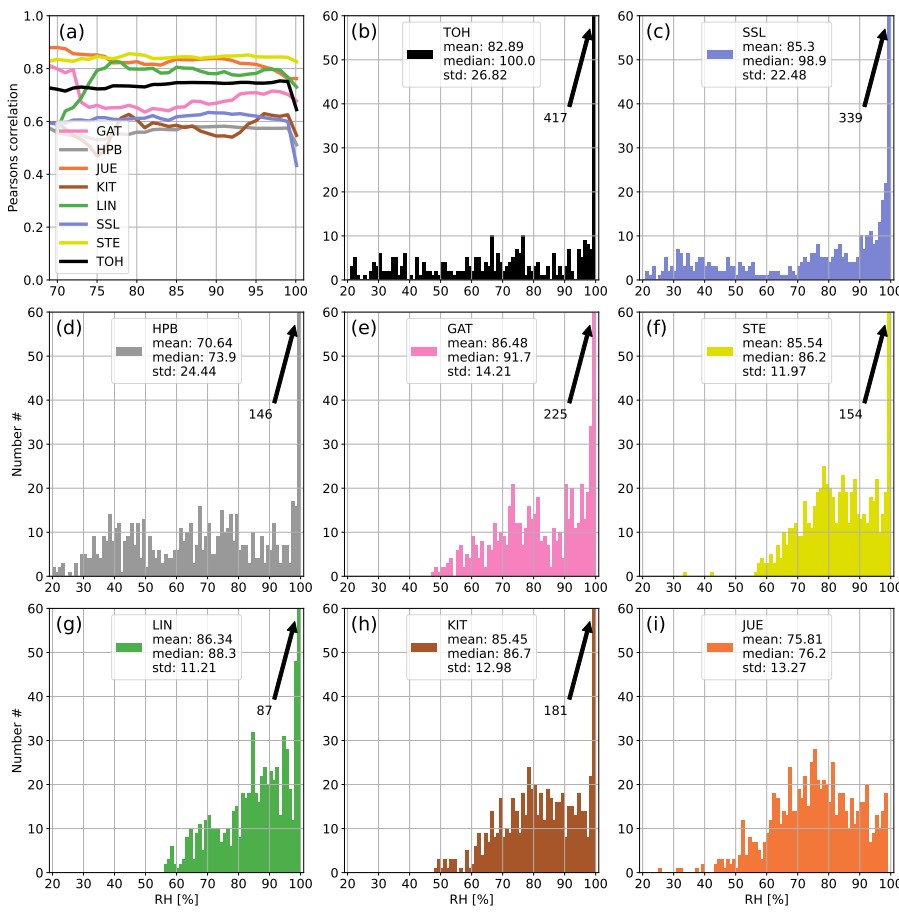

**Figure A11.** The panel (a) shows the development of the correlation coefficient between $^{214}$Po observations and simulated $^{222}$Rn data at all German ICOS stations if data with increasing RH are successively included in the calculation of the correlation. The panels (b) – (i) show the distributions of hourly RH values at the stations of the network. The arrows with numbers indicate values that exceed the y-axis range (usually for high RH). For each station the correlation coefficient was calculated only if the number of filtered data points was higher than 10% of all values for the given month to avoid the stochastic behaviour of the curves in the first subplot associated with small data sets. November 2022.

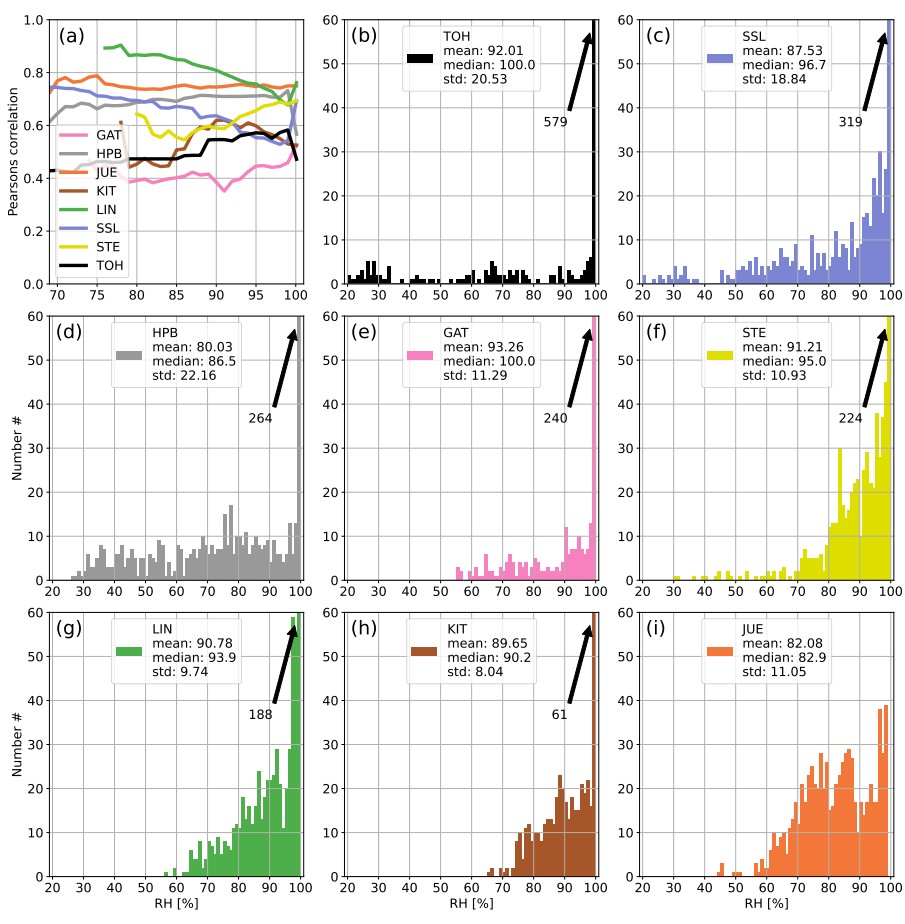

**Figure A12.** The panel (a) shows the development of the correlation coefficient between [214]Po observations and simulated [222]Rn data at all German ICOS stations if data with increasing RH are successively included in the calculation of the correlation. The panels (b) – (i) show the distributions of hourly RH values at the stations of the network. The arrows with numbers indicate values that exceed the y-axis range (usually for high RH). For each station the correlation coefficient was calculated only if the number of filtered data points was higher than 10% of all values for the given month to avoid the stochastic behaviour of the curves in the first subplot associated with small data sets. December 2022.



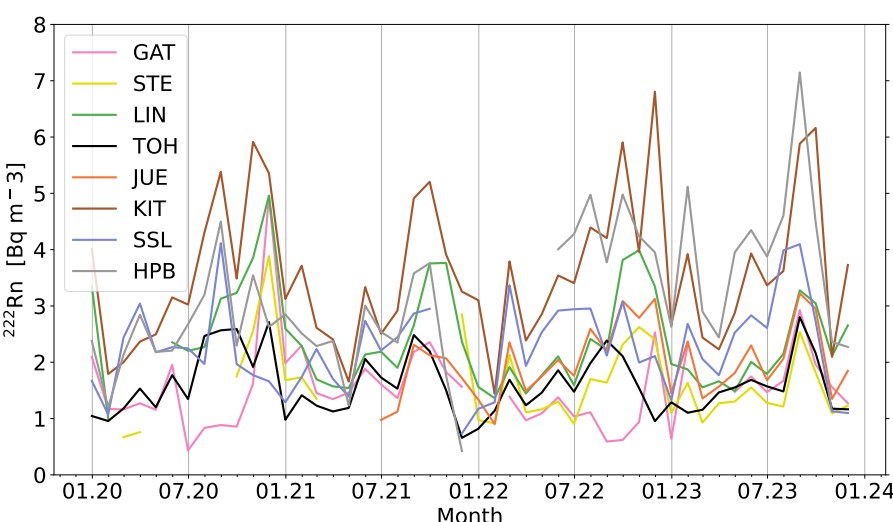

**Figure A13.** Seasonal cycle of $^{222}$Rn monthly means at all German stations in the time period of 2020-2023. Note that monthly values are only shown if more than 70% of the daily mean values were available for the respective month.

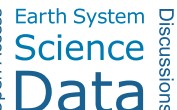

**Table 1.** Parameters and locations of German ICOS stations with $^{214}$Po measurements

| Station | Station code (GAW ID) | Observation start | Lon (°) | Lat (°) | Station height (m a.m.s.l.) | Intake height (m a.g.l.) | Line length (m) |
|---|---|---|---|---|---|---|---|
| Gartow | GAT | November 2015 | 53° 04' N | 11° 26' E | 70 | 132 | 147 |
| Steinkimmen | STE | October 2019 | 53° 03' N | 08° 28' E | 29 | 127 | 137 |
| Lindenberg | LIN | February 2015 | 52° 10' N | 14° 07' E | 73 | 98 | 158 |
| Torfhaus | TOH | April 2018 | 51° 49' N | 10° 32' E | 801 | 110 | 119 |
| Jülich | JUE | August 2021 | 50° 54' N | 06° 25' E | 98 | 120 | 131 |
| Karlsruhe | KIT | June 2017 | 49° 06' N | 08° 26' E | 110 | 100 | 112 |
| Schauinsland | SSL | February 2014 | 47° 55' N | 07° 55' E | 1205 | 12 | 12 |
| Hohenpeißenberg | HPB | October 2014 | 47° 48' N | 11° 01' E | 934 | 93 | 1 |