# Peer review of "Radon-222 monitoring at German ICOS atmosphere stations"

_Earth System Science Data, 2024_

## Referee Comment (RC2)

Scott Chambers, Ph.D., Senior Research Scientist
ANSTO, Environment Research & Technology Group

**10ᵗʰ April 2025**

Giulio G.R. Iovine
Editorial Support Team, ESSD

Dear Giulio,

Thank you for the opportunity to review the manuscript "Radon-222 monitoring at German ICOS atmosphere stations" (essd-2024-551) by Gachkivskyi et al., presently under consideration for publication in Earth System Science Data.

The manuscript summarises a range of caveats that apply to atmospheric radon monitoring conducted by "indirect" (single-filter) Heidelberg Radon Monitors, which are routinely used for radon monitoring at many German ICOS stations, and then focuses specifically on problems associated with high relative humidity conditions (related to aerosol scavenging). A method is proposed to limit influences of high humidity conditions on existing and future radon datasets collected using these monitors, and an updated database is provided where potentially affected data has been flagged out.

For potential users of these datasets in the climate and atmospheric science research communities, I see this improvement in quality of archived radon datasets being of great value and significance, since it is a step closer to harmonising these datasets with radon observations made in other parts of Europe (and the world) using direct radon monitoring techniques. I recommend publication of this manuscript after minor revision. I make some general and specific comments below.

**General**

In the abstract and introduction there is discussion about both the utility of radon to distinguish between oceanic and continental air masses (i.e. as an indicator of "baseline" atmospheric conditions; things that might happen at a remote WMO GAW station), and the utility of radon as a tracer of transport and mixing in the atmospheric boundary layer over land. Given that the focus of the manuscript is on radon observations at ICOS stations, I would encourage the authors to focus more on just discussing radon's suitability and usefulness as a tracer of transport and mixing (including model validation).

I think it would be pertinent to mention in the abstract the intended provision of a RH flagged/corrected 11-year radon dataset from the 8 ICOS stations investigated in this paper.

Is it possible, perhaps in the methods section, to provide an approximate detection limit for the HRM under ideal conditions? (>90m agl, <98% humidity, no line loss correction required).

**Specific**

**L24**: Consider saying "half-life" rather than "half-life time" (throughout the manuscript)

**L25**: Vertical radon gradients that occur in the atmosphere related to radon's half-life are typically between the atmospheric boundary layer (ABL) and the free troposphere. This

characteristic is not important in the context of this study (since everything is occurring within the ABL – or at least beneath the synoptic inversion). Vertical gradients that occur within the ABL are typically related to mixing, alone. Most important here is that (i) radon's half-life (3.8 d) is short compared with synoptic time scales (~4-12 d), so it doesn't accumulate in the atmosphere on greater than synoptic timescales, but (ii) radon's half-life is long compared with mixing timescales in the ABL (~ 1 hour), such that it can roughly be considered a conservative tracer for boundary layer mixing and transport studies.

**L32:** "Owing to its applicability as **an** atmospheric transport tracer, $^{222}$Rn observations are recommended as **a** supplementary ..."

**L42:** Isn't there also a HRM operating at the Amazon Tall Tower Observatory? Or has this now stopped?

**L47:** Related to the Jacobi and Andre (1963) reference, consider mentioning the dependence of radioactive equilibrium on height **and** stability here. The authors make this point later (see line 98). Depending on how well the stability dependence is understood, this may be a subject for future investigation given that some of the German ICOS station observations are made at heights within the stable nocturnal boundary layer (based on the diurnal cycles; Figure 6).

**L104-105 (and L124):** At the time of the mentioned ICPs, there was no traceable method to calibrate 1500L ANSTO radon detectors. Only field calibration (on the sample air flow) was possible, leading to uncertainties in *absolute* calibration of order 4-8% associated with necessary assumptions (Chambers et al 2022; Kikaj et al 2025). So, adjusting to an ANSTO detector-based scale could have been problematic. *Relative* radon concentration changes reported by ANSTO 1500L detectors however are very reliable. Importantly, since the 19ENV01 *traceRadon* Project (Röttger et al. 2021) this calibration uncertainty can be removed with the use of Calibration Transfer Standard Devices (Chambers et al. 2022; Röttger et al. 2025).

**L184:** "... transport **away** from the $^{222}$Rn source ..."

**L187-189:** "... south-north gradient of the $^{222}$Rn  concentration is ..."

While the spatial variability of the radon flux is certainly one driver of observed radon concentrations, air mass time in contact with land *en route* to a site (which changes on synoptic and seasonal timescales), also has a strong influence (since the radon half-life is 3.8 days, on synoptic timescales the air mass can have a strong radon "memory" of surface types it has passed over). See also **L193-194**.

**L203-205:** In winter and early spring, if sites are prone to snow cover or frequent cloud cover, then there is usually much less contrast between daytime and nighttime mixing depths, so diurnal amplitudes of trace gas concentrations can be much smaller.

**L215-217:** It is worth considering here that the strongest (stable nocturnal) inversion conditions occur when there is strong surface cooling. If measuring within the stable nocturnal boundary layer (though mainly if closer to the surface), it may not be uncommon to reach values of quite high relative humidity. I doubt that for the measurements in this study (>90m agl) even if they are within the SNBL (as indicated by diel radon cycles in the warmer months) that high RH values would be common.

**References**

Chambers, S., Griffiths, A., Williams, A., Sisoutham, O., Morosh, V., Röttger, S., Mertes, F., and Röttger, A. Portable two-filter dual-flow-loop $^{222}$Rn detector: stand-alone monitor and calibration transfer device. Advances in Geoscience, 57, 63-80, 2022, https://doi.org/10.5194/adgeo-57-63-2022.

Kikaj D, E Chung, AD Griffiths, SD Chambers, G Forster, A Wenger, P Pickers, C Rennick, S O'Doherty, J Pitt, K Stanley, D Young, LS Fleming, K Adcock, E Safi and T Arnold. Direct high-precision radon quantification for interpreting high-frequency greenhouse gas measurements. Atmos. Meas. Tech., 18, 151–175, 2025, https://doi.org/10.5194/amt-18-151-2025.

Röttger, A., Röttger, S., Grossi, C., Vargas, A., Curcoll, R., Otáhal, P., Hernández-Ceballos, M. A., Cinelli, G., Chambers, S. D., Barbosa, S. A., Ioan, M.-A., Radulescu, I., Kikaj, D., Chung, E., Arnold, T., Yver-Kwok, C., Fuente, M., Mertes, F., and Morosh, V. New metrology for radon at the environmental level. Meas. Sci. Technol., 32, 124008, https://doi.org/10.1088/1361-6501/ac298d, 2021.

Röttger S, A Röttger, F Mertes, S Chambers, A Griffiths, R Curcoll and C Grossi. Traceable low activity concentration calibration of radon detectors for climate change observation networks. Measurement: Sensors, https://doi.org/10.1016/j.measen.2024.101708, 2025.

---

## Author Comment (AC1)

We want to thank Alan Griffiths for the review of our manuscript and his helpful suggestions for improving this paper. Our replies are marked in blue.

**General comments**

This manuscript accompanies a radon-222 dataset (derived from measuring radon's decay product) from eight German monitoring stations with measurements dating back to 2014. Compared with earlier publications, this release handles the effect of very humid conditions on the measurement by flagging out periods when humidity exceeds a particular threshold.

With this additional quality control measure, the data are ready for immediate use in subsequent analyses. In my opinion, because radon-222 is an important natural tracer, and because of the other measurements available from these ICOS stations, this data set is very likely to be used in a range of studies.

The method for determining the humidity threshold is appropriate, the reasoning behind it is transparent, and the one-filter measurement technique is well described in previous publications, which are appropriately cited.

Regarding the linked data set, the data is of high quality, well formatted and well described. Apart from the queries below, which should be simple to address, I consider that the data set will be reused productively in the future and recommend the manuscript for publication.

**Specific comments**

I have three minor suggestions; two related to the humidity threshold and one observation about the data itself.

First, I am uncertain about whether there is a single humidity threshold, applied to all stations, or if the humidity threshold is different for each station ("We have therefore developed relative humidity (RH) flagging thresholds for the individual stations...", line 59). Elsewhere, including the dataset landing page, it is implied that data is flagged as passing manual QC only when RH<98%, which is to say a single threshold of 98% is used across all sites. If this is the case, an unambiguous statement around line 225 (conclusions) and in the abstract is recommended; if the threshold is station-dependent then the threshold (as used during QC of the published data) should be included in Table 1.

It is accurate that certain statements in the paper may lead to the conclusions that different threshold to 98% should be applied to some of the stations or measurement conditions. That was not our intention. During the early stages of the investigation there were some indications that the mountain stations may require a lower RH threshold (95% was discussed), however, as more data became available this assumption was abandoned. The text has now been revised to reflect the final stage of the study (lines 13, 61 and 228).

Second, it seems rather likely that a particular use might require a different humidity threshold. This would be extremely straightforward if the humidity values were included in the data files, or acceptably straightforward if links to the meteorological data were included in this paper (along with instructions about which humidity sensor to use to replicate the published threshold, as there are likely to be many at each site).

The meteorological data used and cited in this publication (line 147) has previously been published elsewhere (TOH, HPB, GAT, STE, LIN, KIT, JUE) or is available upon request from the data providers

(SSL data from UBA, DWD/pre-ICOS LIN, 2015-2017, and HPB, 2014-2015, from). For the ICOS stations, the RH data can be unambiguously associated with the radon data as there is only one RH series per measurement height. For the SSL station we indeed added the humidity sensor designation to the corresponding citation. We added the "citation" and contact persons for the pre-ICOS LIN and HPB RH data. For these reasons, we didn't deem it to be necessary to include humidity data to the radon data set. Furthermore, we want to encourage the data users to use and cite the original meteorological data sets if they want to use them for e.g. applying different RH threshold than suggested in this publication.

Third, there is a period of data at the beginning of the Schauinsland (SSL) record, from February 2014, which is anomalously high compared with the rest of the record even though it is flagged "O" (Manual QC passed).  Since this is at the start of the record, and there is a break in monitoring before 'normal' measurements resume, it seems worth double-checking the classification (or making a note in the paper about what may have caused this – if it is thought to be non-instrumental).

We are grateful for attention being drawn to the fact that the initial period of the SSL data displays anomalous behaviour. Indeed, during this measurement period, the HRM in SSL was not connected to the main intake line and was measuring the room air for the calibration purposes. This period is removed from the updated data release.

Previous version of the data can be assessed with the following link:
https://meta.icos-cp.eu/collections/9wBiDzAqCiELPe2rARbIoggZ

Updated version of the data can be found here:
https://meta.icos-cp.eu/collections/vlcPkY6KbrbIpET1aYq5ps07

Same DOI was assigned to the updated data set and will be published in the paper:
https://doi.org/10.18160/Q2M8-B1HJ

**Technical comments**

Line 13: "..about 98% RH…", if the threshold of 98% was used uniformly across all sites then add a comment here.

A clarification was added to avoid misconception about several RH thresholds: "…about 98% RH and was applied uniformly at all measurement sites…".

L 17: "…flat areas…"  I read this as implying that the mountain sites are not useful (at least, not 'analysis-ready'), even when humidity is low.  I'm not sure that this is intended, based on the rest of the paper. In any case, the abstract should provide concrete guidance, to avoid the misuse of this data set, by linking these recommendations to how a new user could get started.  For instance, a statement like, "Measurements flagged as passing quality control from the stations GAT, STE, LIN, JUE, and KIT meet these criteria whereas other measurements should be treated with more care".

As was correctly pointed out, this was not the intention. The emphasis on the stations in the flat area is due to the significant difference in the high RH values in the flat terrain station in comparison to the mountain stations (e.g. 51% of flagged data in TOH during winter). The conclusion drawn is that the HRM, with its progeny measurement principle, may be more suitable for deployment in flat terrain stations where high RH situations are less prevalent. The section regarding the 'flat areas' was removed from the abstract to avoid confusion (line 17), and the clarification passage was added to the discussion (line 257).

L19: I think that typical style for isotopes, when the element name is written out in full, is the hyphenated form (Radon-222)

Thank you, changes are applied throughout the paper.

L20: "…as gaseous constituent…" -> "as a gas"

Suggested changes are applied.

L51: "function" -> "functions"

The "function" was correct but "were" was changed to "was".

**Dataset**

A typo in data headers ("depent" -> "dependent"):  Disequilibrium: specifies the sampling height-depent factor between calculated atmospheric 222Rn activity concentration in air and measured 214Po activity concentration in air

There is a column called "QualityId" – not defined in the data file headers (if this is of no use to the end-user, it could be described in the headers as "Heidelberg University internal use only")

Thank you for bringing our attention to this. The typo was corrected and additional explanation to the "QualityId" column was added to the updated data set.

There is a header describing the Data Format as Version 1.0.  Is there a link to this format, for example is it standardised across the ICOS network?  If so, is there any sample code in popular analysis languages (R, Python) which would read the data and apply the QC flags?  If sample code like this does exist, it could be linked from the data files or from the ESSD paper.  This is not necessary, as the data file is in a simple plain-text format, but some users may benefit from some demonstration code and therefore be more likely to access the data and use it correctly.

This is a valid point. However, the long-term plan for these HRM radon data is for them to become part of the official ICOS data. As soon as this happens, the files will be 'ingested' into the Carbon Portal data pool. They can then also be read using the icoscp Python library. Therefore, we do not want to create an additional tool that could potentially become obsolete in the near future.

---

## Author Comment (AC2)

We would like to thank Scott Chambers for his review of our manuscript and his constructive suggestions for improving this publication. Responses are indicated in blue.

**Scott Chambers, Ph.D., Senior Research Scientist**

ANSTO, Environment Research & Technology Group

**10th April 2025**

Giulio G.R. Iovine

Editorial Support Team, ESSD

Dear Giulio,

Thank you for the opportunity to review the manuscript "Radon-222 monitoring at German ICOS atmosphere stations" (essd-2024-551) by Gachkivskyi et al., presently under consideration for publication in Earth System Science Data.

The manuscript summarises a range of caveats that apply to atmospheric radon monitoring conducted by "indirect" (single-filter) Heidelberg Radon Monitors, which are routinely used for radon monitoring at many German ICOS stations, and then focuses specifically on problems associated with high relative humidity conditions (related to aerosol scavenging). A method is proposed to limit influences of high humidity conditions on existing and future radon datasets collected using these monitors, and an updated database is provided where potentially affected data has been flagged out.

For potential users of these datasets in the climate and atmospheric science research communities, I see this improvement in quality of archived radon datasets being of great value and significance, since it is a step closer to harmonising these datasets with radon observations made in other parts of Europe (and the world) using direct radon monitoring techniques. I recommend publication of this manuscript after minor revision. I make some general and specific comments below.

**General**

In the abstract and introduction there is discussion about both the utility of radon to distinguish between oceanic and continental air masses (i.e. as an indicator of "baseline" atmospheric conditions; things that might happen at a remote WMO GAW station), and the utility of radon as a tracer of transport and mixing in the atmospheric boundary layer over land. Given that the focus of the manuscript is on radon observations at ICOS stations, I would encourage the authors to focus more on just discussing radon's suitability and usefulness as a tracer of transport and mixing (including model validation).

We would like to thank you for your suggestion. We think that all mentioned applications of radon play an important role in environmental studies and may be of interest to the potential readers. We have provided a brief description of these applications alongside a few examples, without placing additional emphasis on any of them. Furthermore, the focus of this study is not on the applications of radon; rather, the aim of this section is to provide potential readers with notable examples. Therefore, we argue that the abstract and the opening paragraph of this paper should not be altered in this regard.

I think it would be pertinent to mention in the abstract the intended provision of a RH flagged/corrected 11-year radon dataset from the 8 ICOS stations investigated in this paper.

Yes, you are correct. A sentence concerning the dataset was added at the end of the abstract (line 17).

Is it possible, perhaps in the methods section, to provide an approximate detection limit for the HRM under ideal conditions? (>90m agl, <98% humidity, no line loss correction required).

It should be noted that no experiments concerning the detection limit of the HRM have been conducted recently. The detection limit reported for an earlier generation of the HRM is 0.5 Bq/m-3 (Levin, 2002). All comparative studies carried out between the current and previous iterations of HRMs suggest that there are no significant differences between them (Rosenfeld, 2010). However, in Grossi 2020, the detection limit of 0.07 Bq/m-3 was reported. The latter value alongside the corresponding definition will be used in this paper.

**Specific**

**L24:** Consider saying "half-life" rather than "half-life time" (throughout the manuscript)

Suggested changes are applied.

**L25:** Vertical radon gradients that occur in the atmosphere related to radon's half-life are typically between the atmospheric boundary layer (ABL) and the free troposphere. This characteristic is not important in the context of this study (since everything is occurring within the ABL - or at least beneath the synoptic inversion). Vertical gradients that occur within the ABL are typically related to mixing, alone. Most important here is that (i) radon's half-life (3. 8 d) is short compared with synoptic time scales (~4-12 d), so it doesn't accumulate in the atmosphere on greater than synoptic timescales, but (ii) radon's half-life is long compared with mixing timescales in the ABL (~1 hour), such that it can roughly be considered a conservative tracer for boundary layer mixing and transport studies.

Your reasoning is valid. The vertical gradients that we observe and study at the measurement sites in the German ICOS network are caused by vertical mixing, which is part of ABL mixing on an hourly scale. Therefore, the statement in the first paragraph of the introduction that the half-life of Rn is 'short enough' for the observation of vertical mixing is irrelevant in the context of this study. This passage has been reworked according to your comments, and the statement about the suitability of Rn due to its conservative nature compared to ABL mixing timescales has been added (lines 25-33).

**L32:** "Owing to its applicability as **an** atmospheric transport tracer, 222Rn observations are recommended as **a** supplementary..."

Suggested changes are applied.

**L42:** Isn't there also a HRM operating at the Amazon Tall Tower Observatory? Or has this now stopped?

Yes, thank you for pointing this out. We will mention it and add the citation to the newly published ATTO dataset, which was already flagged according to the guidelines in this paper (line 44).

**L47:** Related to the Jacobi and Andre (1963) reference, consider mentioning the dependence of radioactive equilibrium on height **and** stability here. The authors make this point later (see line 98). Depending on how well the stability dependence is understood, this may be a subject for future

investigation given that some of the German ICOS station observations are made at heights within the stable nocturnal boundary layer (based on the diurnal cycles; Figure 6).

The 'atmospheric stability' was incorporated into the line 49. Indeed, this potential study may prove to be of interest in the context of HRM measurement principles. The deployment of a "bias-free" radon monitor (e.g. ANSTO) at potential investigation sites may help to disentangle the disequilibrium factor and its dependence on atmospheric stability.

**L104-105 (and L124):** At the time of the mentioned ICPs, there was no traceable method to calibrate 1500L ANSTO radon detectors. Only field calibration (on the sample air flow) was possible, leading to uncertainties in absolute calibration of order 4-8% associated with necessary assumptions (Chambers et al 2022; Kikaj et al 2025). So, adjusting to an ANSTO detector-based scale could have been problematic. Relative radon concentration changes reported by ANSTO 1500L detectors however are very reliable. Importantly, since the 19ENV01 traceRadon Project (Röttger et al. 2021) this calibration uncertainty can be removed with the use of Calibration Transfer Standard Devices (Chambers et al. 2022; Röttger et al. 2025).

Thank you for providing this valuable information. We have updated the relevant paragraph (lines 107–113) to include details of the ANSTO uncertainties and update readers on the current status of the calibration transfer.

L184: "... transport **away** from the 222Rn source ..."

Suggested changes are applied.

L187-189: "... south-north gradient of the 222Rn  **concentration** is..."

Suggested changes are applied.

While the spatial variability of the radon flux is certainly one driver of observed radon concentrations, air mass time in contact with land en route to a site (which changes on synoptic and seasonal timescales), also has a strong influence (since the radon half-life is 3. 8 days, on synoptic timescales the air mass can have a strong radon "memory" of surface types it has passed over). See also **L193-194.**

We added the comment about potential contribution of the marine air masses from North Sea and Baltic Sea as well as longer air mass residence time over radon emanating soils for the southern stations, which could contribute to the observed radon north-south gradient (lines 198-201).

**L203-205:** In winter and early spring, if sites are prone to snow cover or frequent cloud cover, then there is usually much less contrast between daytime and nighttime mixing depths, so diurnal amplitudes of trace gas concentrations can be much smaller.

This potential explanation together with the citation (Emeis and Turk, 2004) of the mixing height distributions over the course of the year was added to the paragraph (lines 218-220).

 **L215-217:** It is worth considering here that the strongest (stable nocturnal) inversion conditions occur when there is strong surface cooling. If measuring within the stable nocturnal boundary layer (though mainly if closer to the surface), it may not be uncommon to reach values of quite high relative humidity. I doubt that for the measurements in this study (>90m agl) even if they are within the SNBL (as indicated by diel radon cycles in the warmer months) that high RH values would be common.

We concur with the reasoning outlined above; however, we do not consider it necessary to mention the possibility of high RH values due to strong surface cooling, as this is not relevant for the stations in this study (as indicated in your comment).

References

Chambers, S., Griffiths, A., Williams, A., Sisoutham, O., Morosh, V., Röttger, S., Mertes, F., and Röttger, A. Portable two-filter dual-flow-loop 222 Rn detector: stand-alone monitor and calibration transfer device. Advances in Geoscience, 57, 63-80, 2022, https://doi.org/10.5194/adgeo-57-63-2022.

Kikaj D, E Chung, AD Griffiths, SD Chambers, G Forster, A Wenger, P Pickers, C Rennick, S O'Doherty, J Pitt, K Stanley, D Young, LS Fleming, K Adcock, E Safi and T Arnold. Direct high-precision radon quantification for interpreting high-frequency greenhouse gas measurements. Atmos. Meas. Tech., 18, 151-175, 2025, https://doi.org/10.5194/amt-18-151-2025.

Röttger, A., Röttger, S., Grossi, C., Vargas, A., Curcoll, R., Otáhal, P., Hernández-Ceballos, M. A., Cinelli, G., Chambers, S. D., Barbosa, S. A., loan, M.-A., Radulescu, I., Kikaj, D., Chung, E., Arnold, T., Yver-Kwok, C., Fuente, M., Mertes, F., and Morosh, V. New metrology for radon at the environmental level. Meas. Sci. Technol., 32, 124008, https://doi.org/10.1088/1361-6501/ac298d, 2021.

Röttger S, A Röttger, F Mertes, S Chambers, A Griffiths, R Curcoll and C Grossi. Traceable low activity concentration calibration of radon detectors for climate change observation networks. Measurement: Sensors, https://doi.org/10.1016/j.measen.2024.101708, 2025.

Emeis, Stefan, and Matthiast Turk. "Frequency distributions of the mixing height over an urban area from SODAR data." *Meteorologische Zeitschrift* 13.5 (2004): 361-368.

Levin, I., Born, M., Cuntz, M., Langendörfer, U., Mantsch, S., Naegler, T., Schmidt, M., Varlagin, A., Verclas, S., and Wagenbach, D.: Observations of atmospheric variability and soil exhalation rate of radon-222 at a Russian forest site. Technical approach and deployment for boundary layer studies, Tellus B: Chemical and Physical Meteorology, 54, 462–475, 2002.

Rosenfeld, M.: Modifikation des Heidelberger Radon-Monitors und erste Messungen, Diploma Thesis, University of Heidelberg, Heidelberg, Germany, 2010.

Grossi, C., Chambers, S. D., Llido, O., Vogel, F. R., Kazan, V., Capuana, A., Werczynski, S., Curcoll, R., Delmotte, M., Vargas, A., et al.: Intercomparison study of atmospheric 222 Rn and 222 Rn progeny monitors, Atmospheric Measurement Techniques, 13, 2241‑2255, 2020.